# Synergistic Catalysis of SnO_2_/Reduced Graphene Oxide for VO^2+^/VO_2_^+^ and V^2+^/V^3+^ Redox Reactions

**DOI:** 10.3390/molecules26165085

**Published:** 2021-08-22

**Authors:** Yongguang Liu, Yingqiao Jiang, Yanrong Lv, Zhangxing He, Lei Dai, Ling Wang

**Affiliations:** School of Chemical Engineering, North China University of Science and Technology, Tangshan 063009, China; hglyg@ncst.edu.cn (Y.L.); jiangying_qiao@163.com (Y.J.); yronglv@163.com (Y.L.); dailei_b@163.com (L.D.)

**Keywords:** vanadium redox reaction, electrochemical kinetics, SnO_2_/reduced graphene oxide, catalyst

## Abstract

In spite of their low cost, high activity, and diversity, metal oxide catalysts have not been widely applied in vanadium redox reactions due to their poor conductivity and low surface area. Herein, SnO_2_/reduced graphene oxide (SnO_2_/rGO) composite was prepared by a sol–gel method followed by high-temperature carbonization. SnO_2_/rGO shows better electrochemical catalysis for both redox reactions of VO^2+^/VO_2_^+^ and V^2+^/V^3+^ couples as compared to SnO_2_ and graphene oxide. This is attributed to the fact that reduced graphene oxide is employed as carbon support featuring excellent conductivity and a large surface area, which offers fast electron transfer and a large reaction place towards vanadium redox reaction. Moreover, SnO_2_ has excellent electrochemical activity and wettability, which also boost the electrochemical kinetics of redox reaction. In brief, the electrochemical properties for vanadium redox reactions are boosted in terms of diffusion, charge transfer, and electron transport processes systematically. Next, SnO_2_/rGO can increase the energy storage performance of cells, including higher discharge electrolyte utilization and lower electrochemical polarization. At 150 mA cm^−2^, the energy efficiency of a modified cell is 69.8%, which is increased by 5.7% compared with a pristine one. This work provides a promising method to develop composite catalysts of carbon materials and metal oxide for vanadium redox reactions.

## 1. Introduction

With the development of industrial processes, energy consumption is increasing day by day. However, energy consumption brings environmental pollution [1,2,3,4]. In order to solve this problem and achieve “carbon neutrality”, humankind began to explore renewable energy [5,6,7,8,9]. However, this type of energy is not always available at all times, and in most cases, due to the oscillation of energy availability in its cycle and peak demand, supply and demand are not synchronized [10,11,12,13,14,15]. Energy storage equipment is needed to store and transform the electric energy generated by renewable energy sources to realize its large-scale application [16,17,18,19,20]. Among many energy storage devices, a vanadium redox flow battery (VRFB) has attracted much attention because of its unique performance [21,22,23]. The main components of a VRFB are electrodes, electrolytes, and proton exchange membranes [24,25,26]. The performance of an electrode has a direct impact on the overall performance of the battery. Carbon-based materials have become the most widely used electrode material in a VRFB due to their large specific surface area and good corrosion resistance. However, the electrochemical performance of carbon-based materials is poor, which is not conducive to their large-scale utilization. In order to improve the performance of carbon-based materials and expand the use of materials, researchers have modified them. Common electrode modification methods mainly include intrinsic treatment and the introduction of catalysts.

There are several categories of catalysts, including carbon materials, metal, metal oxide, metal nitride, metal boride, and metal carbide. Firstly, carbon materials (CNTs, graphene, carbon sheet, carbon nanofiber, etc.) have advantages including a low cost, obvious structure, and a larger surface area, and have been widely used as a catalyst in VRFB [27,28,29]. For example, Yan et al. [30] developed multi-walled carbon nanotubes as a catalyst for VO^2+^/VO_2_^+^ redox reaction. In addition, some biomass-based carbon materials derived from glucose, fish scales, and kiwi fruit have abundant groups, heteroatoms, and a large surface area, which also show certain electrochemical catalysts for a vanadium redox reaction [31,32]. Metal has a unique atomic structure and high conductivity, and exhibits intrinsic catalysis for a vanadium redox reaction. Metal catalysts mainly include Au, Pt, Ir, Cu, etc. Zeng et al. [33] used a copper nanoparticle to decorate graphite felt with in situ electrodeposition. The copper nanoparticle was an excellent catalyst for a V^3+^/V^2+^ redox reaction, and improved the energy efficiency of the VRFB. However, most of the metal catalysts have disadvantages, including a high price and easy access to hydrogen evolution. Moreover, metal oxides have attracted more attention due to the advantages of a low cost, high stability, and diversity. In recent decades, many metal oxides including ZrO_2_, TiO_2_, CeO_2_, WO_3_, SnO_2,_ etc_.,_ have been developed and applied in VRFB [34,35,36,37,38,39]. For instance, Chen et al. [35] modified graphite felt with a ZrO_2_ nanoparticle by thermal decomposition. ZrO_2_ showed good wettability and acted as active sites for both redox reactions, resulting in faster mass and charge transfer processes. However, due to the low conductivity, the intrinsic catalysis of metal oxide is not efficiently used. Other metal compounds have been developed as catalysts for VRFB. For example, the Ti_3_C_2_T_x_ MXene spheres have metallic behavior and a high electrical conductivity, which shows high catalysis for a V^3+^/V^2+^ redox reaction [40]. Moreover, metal borides such as TiB_2_ and ZrB_2_ exhibit high conductivity for a V^3+^/V^2+^ redox reaction [41,42]. It is noteworthy that carbon materials can be used as support to not only improve conductivity of metal oxide but also to increase the surface area of metal oxide. Therefore, it is an efficient way to obtain high-performance catalysts by compositing carbon materials and metal oxide. For example, Li et al. [43] synthesized a polydopamine-Mn_3_O_4_ composite as a catalyst for a VO^2+^/VO_2_^+^ redox reaction. The synergistic effect of polydopamine and Mn_3_O_4_ improved the electrochemical performance of the VRFB. 

SnO_2_ is an amphoteric oxide, which can exist stably in the strong acidic environment of a VRFB electrolyte [39,44]. Due to the wide band gap and unique performance of SnO_2_, it is widely used in optics, sensors, and other fields [45]. A large number of experiments have proved that SnO_2_ has a good catalytic effect on V^3+^/V^2+^ and VO_2_^+^/VO^2+^ redox reactions. Ha et al. [39] prepared in situ SnO_2_ nanoparticles and modified carbon felt electrodes by the hydrothermal method. SnO_2_ reduced the activation potential of the reaction and increased the reaction rate. The SnO_2_ modified electrode significantly improves the stability of the battery, which is very important for the VRFB. Graphene has good electrical conductivity, high stability, and a large surface area, making it an excellent electrocatalyst carbon support. Walsh et al. [46] studied the synergy between graphene and Mn_3_O_4_. Due to the interaction between graphene support and the Mn_3_O_4_ catalyst, the catalytic performance of the composite material is higher than that of any component alone. 

In this article, we used SnO_2_ as the electrocatalyst and reduced graphene oxide as the carbon support, respectively. The large surface area of reduced graphene oxide not only makes the SnO_2_ nanoparticle more dispersive, but also provides a larger reaction place for redox reaction. In addition, the high conductivity of reduced graphene oxide boosts the transport of electrons for redox reaction. Therefore, it is inferred that the synergetic effect between reduced graphene oxide and SnO_2_ is more conducive to a vanadium redox reaction. The catalytic performance of the SnO_2_/reduced graphene oxide composite catalyst for V^3+^/V^2+^ and VO_2_^+^/VO^2+^ reactions is much higher than using SnO_2_ and reduced graphene oxide alone as a catalyst. Due to the synergy between SnO_2_ and reduced graphene oxide, the composite catalyst has good electrochemical activity and stability.

## 2. Experimental

### 2.1. Preparation of Materials

As shown in Figure 1, the sample was prepared by the sol–gel method. First, 1.4977 g of SnCl_2_·2H_2_O was weighed and placed in a beaker. Then, 13.27 mL of an anhydrous ethanol/water mixed solution (volume ratio: 15:1) was added and stirred in a water bath at 65 °C for 40 min to gradually form a gel. After the gel was formed, it was aged in air for 24 h to obtain a precursor of tin sol. Then, 50 mg of graphene oxide (GO), the tin sol precursor, and 10 mL of anhydrous alcohol were mixed and ultrasonically dispersed for 1 h. Next, the sample was placed in a drying oven to dry adequately at 80 °C. The Ar was put into the tube furnace before calcining for 2 h to exhaust the air. Then, the sample was calcined at 500 °C for 2 h. The obtained sample was named as SnO_2_/rGO. 

### 2.2. Characterization of Materials

The crystal phases of the samples were studied using the D8 Advance A25 Instrument (Bruker, Berlin, Germany). The microscopic morphology of the sample was observed by a scanning electron microscope (SEM, JSM-IT100). The internal morphology and crystal lattice of the sample were observed by a transmission electron microscope (TEM, JEOL JEM-2100F). The 3H-2000PM1 specific surface area and porous analyzer was used to analyze the materials surface area. The X-ray photoelectron spectroscopy (XPS) of the samples was studied using the Thermo Scientific Escalab 250Xi instrument (Thermo Fisher Scientific, Waltham, MA, USA) to determine the elemental composition of the samples.

### 2.3. Electrochemical Measurements

First, 10 mg of the catalyst was thoroughly mixed with 5 mL of N,N-dimethylformamide (DMF) to obtain a dispersion. The dispersion solution was then put into an ultrasonic cleaning machine and dispersed continuously for 3 h, until the catalyst was evenly dispersed in DMF. The dispersion was then dropped on the glassy carbon electrode (GCE) and left to stand at room temperature for 4 h.

Cyclic voltammetry (CV) and electrochemical impedance spectroscopy (EIS) measurements were performed by an electrochemical workstation (CHI660E, Shanghai Chenghua, Shanghai, China). In this three-electrode system, platinum electrode (1 × 1 cm^2^) was the counter electrode, glassy carbon electrode was the working electrode, and the calomel electrode was the reference electrode. Positive and negative CV and EIS tests were accomplished in 1.6 M VO^2+^ + 3.0 M H_2_SO_4_ and 1.6 M V^3+^ + 3.0 M H_2_SO_4_, respectively. The voltage scanning range of the CV test for a V^2+^/V^3+^ reaction was from −0.75 to −0.1 V, and the VO^2+^/VO_2_^+^ reaction was from 0.2 to 1.5 V. The frequency range for EIS was 1 to 1 × 10^6^ Hz. The EIS of the VO^2+^/VO_2_^+^ and V^2+^/V^3+^ reactions were tested at the polarization of 0.85 V and −0.45 V, respectively.

### 2.4. Charge–Discharge Tests

The catalyst modification cell and the pristine cell were assembled to compare the VRFB’s performance. CT2001A was used for charge–discharge tests. Graphite felt (3 × 3 cm^2^) was activated in a muffle furnace at 400 °C for 10 h. The activated graphite felt was then dipped into the dispersion solution. The dispersion solution was composed of 3 mg of the catalyst and 10 mL of DMF. After the graphite felt completely absorbed the dispersion solution, it was taken out and dried at 80 °C. The above steps were repeated until the dispersion solution was completely absorbed in order to obtain the SnO_2_/rGO modify graphite felt electrode.

The SnO_2_/rGO modify graphite felt was, respectively, used as the positive and negative electrodes of the modified cell. The active graphite felt was used as the pristine cell. The graphite felt was placed in the electrolyte of 0.8 M V^3+^ + 0.8 M VO^2+^ + 3.0 M H_2_SO_4_ to fully absorb the electrolyte. The rate performance was tested at 50, 75, 100, 125, and 150 mA cm^−2^ current densities.

## 3. Results and Discussion

Figure 2a–c show the SEM images of GO, SnO_2_, and SnO_2_/rGO, respectively. As shown in Figure 2a, GO is curly. Figure 2b shows the SnO_2_ particles have obvious agglomeration. It can be seen from Figure 2c that SnO_2_ is evenly distributed on the reduced graphene oxide sheet. As shown in Figure 2d–f, the morphology of the SnO_2_/rGO composite was further characterized by TEM. The low magnification TEM figure shows that a large amount of SnO_2_ is loaded on the reduced graphene oxide sheet. There is a large amount of aggregation, which can be seen more clearly from the medium magnification TEM figure. From the high magnification TEM figure, it can be further concluded that the obvious lattice fringes are 0.34 nm, which is the (110) surface of SnO_2_. Figure 2g, h display the EDX spectra of GO and SnO_2_/rGO, respectively. Absolutely, SnO_2_/rGO have C and O. Compared with Figure 2g, there is Sn in Figure 2h. It can be concluded that there is Sn in SnO_2_/rGO.

Figure 2i shows the X-ray diffraction pattern of the samples. The crystal structure of the samples was studied by XRD. There is a characteristic peak at 12.7° for graphene oxide, which proves that the purity of graphene oxide is very high [47]. The observed pattern of SnO_2_ is consistent with the standard value (JCPDS: 00-021-1250), and there is no characteristic peak of impurity. The reduced graphene oxide crystallization peaks in the XRD pattern of SnO_2_/rGO are lack. The order of the reduced graphene oxide plane is disrupted by the SnO_2_ particles, or reduced graphene oxide sheets are inserted into the SnO_2_ lattice [48]. SnO_2_/rGO has the advantages of two materials. It has both the conductivity of rGO and the catalytic performance of SnO_2_, which can improve the electrochemical catalytic performance.

Figure 3a, b show the N_2_ adsorption and desorption isotherms of GO and SnO_2_/rGO. The isotherms of GO and SnO_2_/rGO are type IV, indicating that the two are mesopores materials. Additionally, the two isotherms exhibit type H3 loops, which identified with plate-like porous aggregates, as these originate from slit-shaped interlayer pores. Based on the multilayer adsorption theory, the specific surface areas of GO and SnO_2_/rGO were computed to be 44.3 m^2^ g^−1^ and 208.9 m^2^ g^−1^, respectively. Pore distribution has been assessed by means of the Barrett–Joyner–Halenda (BJH) method, and the results have been shown in Figure 3c,d. It can be seen that GO and SnO_2_/rGO contain a large number of mesoporous of ~4 nm. The pore width is not affected by heat treatment. The increase of mesoporous can explain why SnO_2_/rGO has a larger surface area, which can provide a larger reaction place for electrode reaction. 

The element composition of SnO_2_/rGO was further analyzed by XPS. The XPS spectrum survey of SnO_2_/rGO is shown in Figure 4a. It shows that the peaks in the energy spectrum of SnO_2_/rGO correspond to C 1s, O 1s, and Sn 3d, indicating that SnO_2_/rGO consists of C, O, and Sn elements. Figure 4b–d show the high-resolution XPS spectra of C 1s, O 1s, and Sn 3d, respectively. There are three peaks centered at 284.4, 285.6, and 288.6 eV, which correspond to C-C, C-O, and O-C=O bonds, respectively. These C 1s peaks related to oxygen-containing groups are very weak. The peaks at 531.1 and 532.3 eV shown in Figure 4c correspond to Sn-O and C-O bonds. These binding energies are characteristic of oxygen ionization on the SnO_2_ surface. Figure 4d shows two different peaks with binding energies of 487.0 and 495.4 eV, corresponding to Sn 3d_5/2_ and Sn 3d_3/2_, respectively. This indicates the presence of Sn^4+^ and the formation of SnO_2_ [49]. It can be concluded that SnO_2_/rGO is composed of reduced graphene oxide and SnO_2_.

CV measurements were used to evaluate the effect of the samples on the electrocatalysis of a VO_2_^+^/VO^2+^ redox reaction. Figure 5a shows the CV curve of bare GCE. It indicates bare GCE without a catalysis for a VO_2_^+^/VO^2+^ redox reaction. It is beneficial to compare the electrochemical activity of the catalysts. The poor peak shape of SnO_2_ is due to the poor electrical conductivity of SnO_2_, which reduces the electrochemical reversibility of the electrode. The peak shape of GO is better than that of SnO_2_, indicating the advantages of GO. The electrochemical performance of SnO_2_/rGO composite is better than those of both GO and SnO_2_. The composite has the synergistic effect of the good electrical conductivity of rGO and the good catalytic property of SnO_2_. Moreover, it can be seen that the peak current of the SnO_2_/rGO electrode (oxidation: 2.59 mA, reduction: 1.19 mA) is the largest, which again shows that SnO_2_/rGO has a good electrochemical activity for a VO_2_^+^/VO^2+^ redox reaction.

Figure 5b,c show the CVs of GO and SnO_2_/rGO. As shown in the figure, with the change of the scan rate, the redox peak current also moves up and down. At a larger scan rate, the difference between the oxidation peak potential and the reduction peak potential gradually increases. Figure 5d depicts the relationship between the redox peak current and the square root of the scan rate. It is clear that there is a proportional relationship between the peak current and the square root of the scan rate, indicating that the reaction is controlled by the diffusion process [50]. The slope of SnO_2_/rGO is higher than that of GO, which may be due to the fact that SnO_2_/rGO has a better catalytic effect, so that the concentration difference on the electrode surface is larger.

To further explore the electrocatalytic property of the samples for redox reaction, we performed an EIS test. Figure 5e displays the Nyquist plots of the positive electrodes of SnO_2_, GO, and SnO_2_/rGO. In the Nyquist diagram, a semicircle appears at the high frequency region and a line appears at the low frequency region. This means that the VO_2_^+^/VO^2+^ electrode reaction is jointly governed by charge and mass transfer processes [51,52]. Figure 5f shows a simplified equivalent circuit diagram conforming to a Nyquist diagram. 

The corresponding fitting electrochemical parameters of three samples are shown in Table 1. The ohmic resistance (R_s_) of SnO_2_/rGO and GO is smaller than that of SnO_2_. In addition, the charge transfer resistance (R_ct_) indicates that SnO_2_/rGO < GO < SnO_2_, which means that SnO_2_/rGO has the lowest R_ct_. The order of electrical double layer capacitance (Q_m_) is SnO_2_ < SnO_2_/rGO < GO. GO has the highest Q_m_ because of a large number of hydrophilic groups on its surface. However, after calcination at high temperature, the hydrophilic groups of the composite are reduced and the Q_m_ is reduced. SnO_2_ has the lowest Q_m_ because it has no hydrophilic groups on its surface. The order of diffusion capacitance (Q_t_) is SnO_2_ < GO < SnO_2_/rGO. This may be due to the comprehensive effect of electrocatalysis and surface changes. The increase of Q_m_ can promote the charge migration of a VO_2_^+^/VO^2+^ reaction, and the increase of Q_t_ can accelerate the diffusion of VO_2_^+^/VO^2+^ ions on the electrode.

To explore the influence of the samples on the electrocatalysis of the V^2+^/V^3+^ reaction, we carried out a CV test. Figure 6a displays the CV curves of GO and SnO_2_/rGO in the 1.6 M V^3+^ + 3.0 M H_2_SO_4_, which indicate that SnO_2_/rGO has a higher redox peak current than GO. The oxidation peak current of SnO_2_/rGO can reach 1.56 mA, which is more than twice as high as that of GO (0.73 mA). The reduction peak current of SnO_2_/rGO (2.82 mA) is 1.55 times higher than that of GO (1.82 mA). The results indicate that SnO_2_/rGO also has superior electrocatalysis towards a V^2+^/V^3+^ redox reaction. This is consistent with the results of the positive electrode. The EIS measurement for the negative couple was carried out to study the electrocatalytic activity. Figure 6b is the Nyquist diagram of the V^2+^/V^3+^ reaction process of GO and SnO_2_/rGO. It can be seen that the Nyquist diagram is similar to a semicircle at the high frequency region and a line in the low frequency region. This means the V^2+^/V^3+^ reaction process is under co-control of the charge and mass transfer processes. Compared with GO, the smaller semicircle and higher slope of SnO_2_/rGO mean lower charge transfer resistance and faster diffusion performance, which comes from the special structure of the SnO_2_/rGO composite. It is the same as the positive EIS measurement analysis results.

Figure 7 shows the SEM images of pristine graphite felt and graphite felt modified with SnO_2_/rGO. It is seen from Figure 7a,b that pristine graphite felt presents a clean and smooth surface. The diameter of the carbon fiber is 10–15 μm. It is observed in Figure 7c,d that graphite felt is uniformly coated with the SnO_2_/rGO catalyst, which offers a larger reaction place and active sites for electrode reaction.

Figure 8a displays the discharge capacity of the pristine and SnO_2_/rGO modified cells at a different current density. It is observed that as the current density increases, the difference in discharge capacity between the two cells progressively increases. This result reflects that SnO_2_/rGO has a good effect on increasing discharge capacity at a high current density. As a result, SnO_2_/rGO can increase the reaction place and active site for electrode reaction, which increases the electrolyte utilization. In Figure 8b, the SnO_2_/rGO cell exhibits a slightly lower than current efficiency (CE) as compared to the pristine cell. This is owing to a longer charge–discharge time of the SnO_2_/rGO cell, and a more severe active substance penetration between two electrolytes in the SnO_2_/rGO cell. In Figure 8c, there is an inversely proportional relation between voltage efficiency (VE) and current density. VE can reflect the cell polarization, including ohmic polarization, electrochemical polarization, and concentration polarization. SnO_2_/rGO can reduce the electrochemical polarization and concentration polarization of the cell. The ohmic resistance of the two cells is comparable, as those two cells are comprised of the same components. The cell with SnO_2_/rGO has a higher VE than the pristine one at 50–150 mA cm^−2^. At 150 mA cm^−2^, the modified cell shows an increase of 6.2% in VE compared to the pristine one (65.7%). This suggests that the SnO_2_/rGO cell exhibits lower electrochemical polarization than the pristine one. As shown in Figure 8d, the energy efficiency (EE) is jointly affected by CE and VE. As the current density rises, the EE of both cells reduces. The SnO_2_/rGO cell presents a greater EE than the pristine one. This indicates that the cell using SnO_2_/rGO has a higher energy storage capacity. At 150 mA cm^−2^, the EE of the modified cell is 69.8%, which is higher than that of the pristine cell (64.1%). These results indicate that the SnO_2_/rGO modified cells can reduce the electrochemical polarization and have better energy storage performance. Figure 8e, f display the charge–discharge curves of two cells. It is seen from the comparison that the curves of the SnO_2_/rGO modified cell exhibit a lower charge voltage and a higher discharge voltage than those of the pristine one. The reason is that the SnO_2_/rGO composite material significantly boosts the redox reaction and reduces the polarization of the cell, suggesting that the SnO_2_/rGO composite raises the energy density of the cell.

## 4. Conclusions

In this work, an easy sol–gel approach was employed to a composite reduced graphene oxide, and SnO_2_ was used as a bifunctional synergistic catalyst. The reduced graphene oxide as carbon support provides an adequate reaction place and fast electron transport due to its large surface area and fine conductivity. Moreover, SnO_2_ has the intrinsic catalysis for a vanadium redox reaction. The strong interaction between the reduced graphene oxide and SnO_2_ results in synergistical catalysis for redox reactions of VO_2_^+^/VO^2+^ and V^2+^/V^3+^, which systematically boosts the electrode reaction from diffusion, charge transfer, and electron transmission processes. Furthermore, SnO_2_/rGO increases the electrolyte utilization of the cell and decreases the electrochemical polarization, resulting in higher discharge capacity and energy efficiency of the modified cell. At 150 mA cm^−2^, the energy efficiency of the modified cell is increased to 69.8%, as compared to 64.1% in the pristine one. 

## Figures and Tables

**Figure 1 molecules-26-05085-f001:**
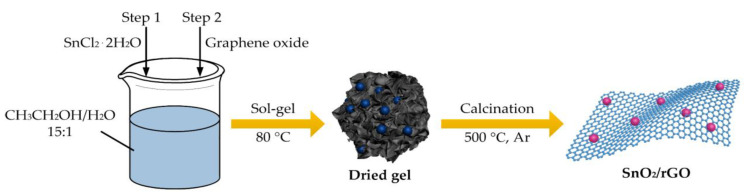
Illustration of preparation procedure of SnO_2_/rGO by sol-gel method.

**Figure 2 molecules-26-05085-f002:**
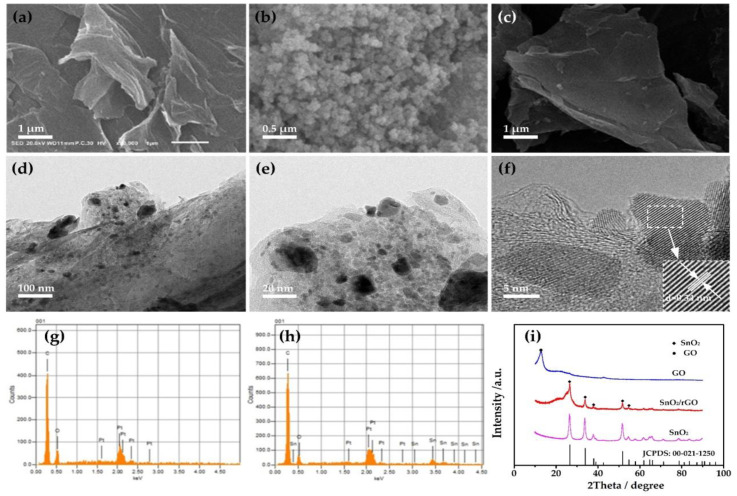
(**a**) SEM image of GO; (**b**) SEM image of SnO_2_; (**c**) SEM image of SnO_2_/rGO; (**d**,**e**) TEM images; (**f**) high-resolution TEM image of SnO_2_/rGO; (**g**) EDX spectra of GO; (**h**) EDX spectra of SnO_2_/rGO; (**i**) XRD patterns of all samples.

**Figure 3 molecules-26-05085-f003:**
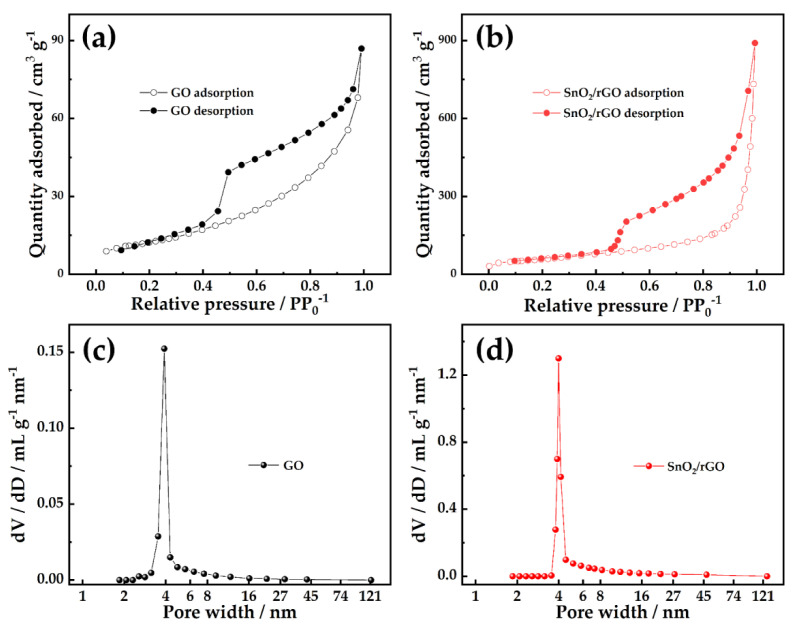
(**a**) N_2_ adsorption–desorption isotherms of GO; (**b**) SnO_2_/rGO; (**c**) pore-size distributions of GO; (**d**) SnO_2_/rGO based on BJH model.

**Figure 4 molecules-26-05085-f004:**
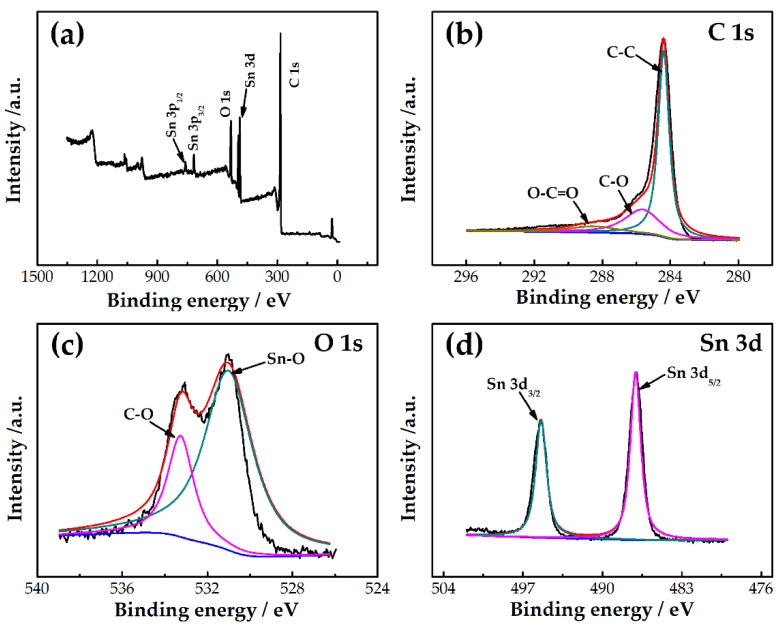
(**a**) XPS spectra of SnO_2_/rGO, and the high-resolution XPS spectra of (**b**) C 1s, (**c**) O 1s, and (**d**) Sn 3d.

**Figure 5 molecules-26-05085-f005:**
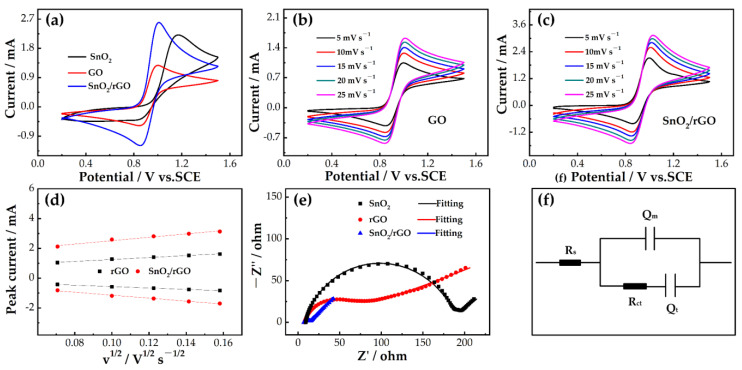
(**a**) CV curves for SnO_2_, GO, and SnO_2_/rGO in 1.6 M VO^2+^ + 3.0 M H_2_SO_4_ at a scan rate of 10 mV s^−1^; (**b**) CVs for GO; (**c**) CVs for SnO_2_/rGO; (**d**) CVs at different scan rates (5–25 mV s^−1^) in 1.6 M VO^2+^ + 3.0 M H_2_SO_4_, and plots of the redox peak current versus the square root of the scan rate for GO and SnO_2_/rGO electrodes; (**e**) Nyquist plots for SnO_2_, GO, and SnO_2_/rGO in 1.6 M VO^2+^ + 3.0 M H_2_SO_4_ at 0.85 V; (**f**) simplified electrical equivalent circuit fitting with Nyquist plots.

**Figure 6 molecules-26-05085-f006:**
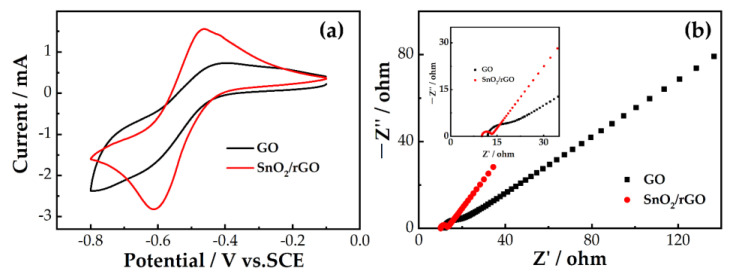
(**a**) CV curves; (**b**) CV curves at the scan rate of 10 mV s^−^^1^ and Nyquist plots for GO and SnO_2_/rGO in 1.6 M V^3+^ + 3.0 M H_2_SO_4_ electrolyte.

**Figure 7 molecules-26-05085-f007:**
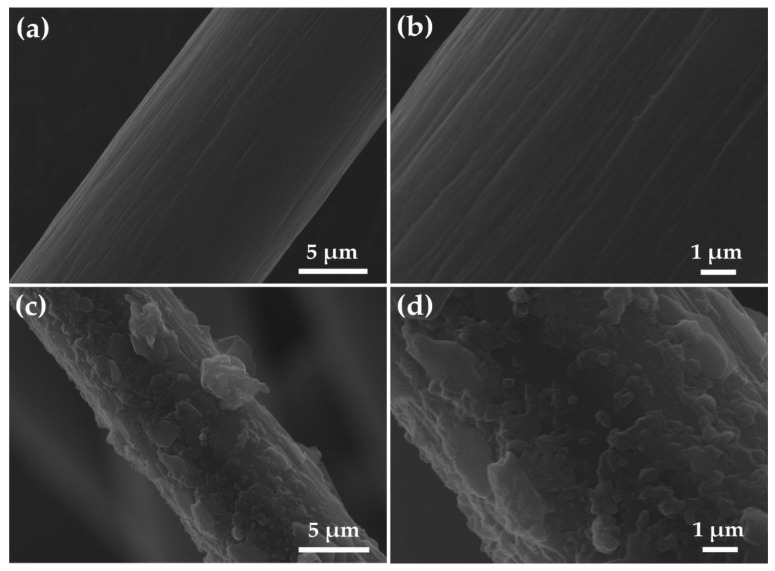
(**a**,**b**) SEM images of pristine graphite felt; (**c**,**d**) modified graphite felt with SnO_2_/rGO.

**Figure 8 molecules-26-05085-f008:**
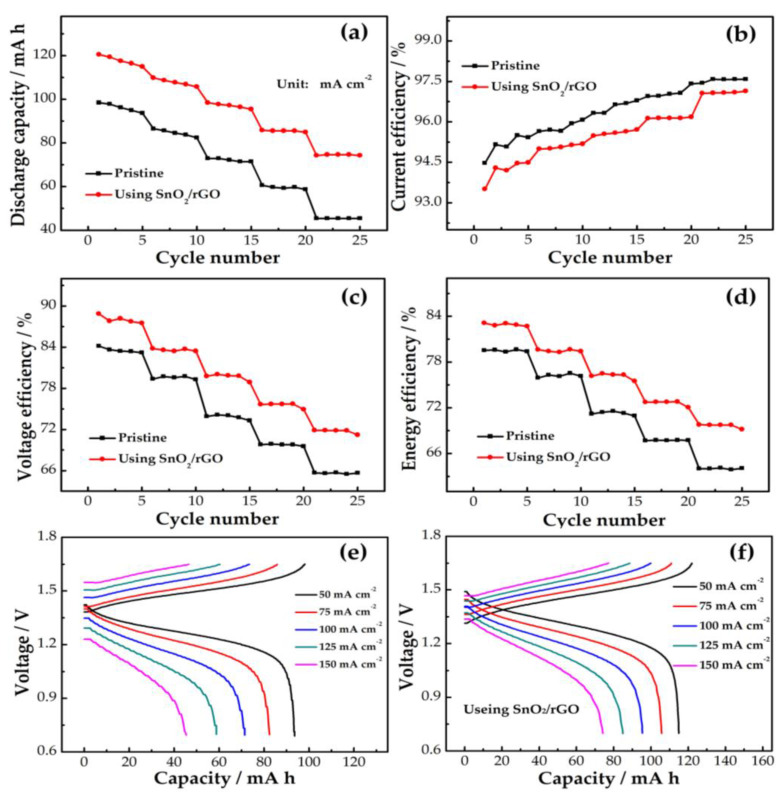
(**a**) Discharge capacity; (**b**) CE, (**c**) VE, and (**d**) EE of cells with and without SnO_2_/rGO; (**e**) charge–discharge curves of pristine cell; (**f**) SnO_2_/rGO modified cell at current density of 50–150 mA cm^−2^.

**Table 1 molecules-26-05085-t001:** Corresponding fitting electrochemical parameters for SnO_2_, GO, and SnO_2_/rGO.

Sample	R_s_/Ω	Q_m_		R_ct_/Ω	Q_t_	
		Y_0_	n_0_		Y_1_	n_1_
GO	7.771	6.43 × 10^−3^	0.24	37.4	2.86 × 10^−5^	0.859
SnO_2_	8.947	3.12 × 10^−2^	0.48	177.5	3.08 × 10^−5^	0.856
SnO_2_/rGO	7.001	3.14 × 10^−2^	0.53	7.41	4.95 × 10^−4^	0.663

## Data Availability

Not applicable.

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
