# Peer review of "Synergistic Catalysis of SnO2/Reduced Graphene Oxide for VO2+/VO2+ and V2+/V3+ Redox Reactions"

_molecules, 2021, doi:10.3390/molecules26165085_

Round 1

Reviewer 1 Report

the authors report an improved cycling performances with GF modified with reduced graphene/SnO2 composite. The authors claim some synergetic effect.

It is hardly clear why SnO2/reduced graphene would improve the electrocatalytic properties of the GF electrode in the battery.

What about the surface developed by the rGO and rGO/SnO2? It is the reason for the increased current on glassy carbon modified by these materials. Surface area is often used in the discussion but no quantitative data are provided.

Is missing some caracterization of the modified felt. Is it homogenous modification.

Pb with plot of impedance spectra that must show identical x and y axis.

It is truly graphene? It looks like relatively thick material on the SEM images.

What about the stability of this composite material in the conditions of VRFB?

Overall, the authors claims for improved performances. They must improve the discussion about the mechanism that would allow any improvment compare to an activated GF.

I do not expect a high impact of this work in the field of RFB.

Author Response

Response to reviewers for Manuscript

Title: Synergistic catalysis of SnO2/reduced graphene oxide for VO2+/VO2+ and V2+/V3+ redox reactions

Corresponding Author: Mr. Zhangxing He

All Authors: Yongguang Liu, Yingqiao Jiang, Yanrong Lv, Zhangxing He, Lei Dai, Ling Wang.

Dear editor,

We must thank you and the reviewer for the hard work. Please do forward our heartfelt thanks to the expert for the valuable comments. Based on the comments and suggestions we received, we have made careful modifications and supplemented data on the previous manuscript. All changes are marked in red in the paper. We hope the revised manuscript will meet your magazine’s standard and the review’s requirements. Below you will find our point-by-point responses to the reviewer’s comments:

 The authors report an improved cycling performances with GF modified with reduced graphene/SnO2 composite. The authors claim some synergetic effect.

  1. It is hardly clear why SnO2/reduced graphene would improve the electrocatalytic properties of the GF electrode in the battery.

Reply: Thank you. SnO2/rGO, as a catalyst, is supported on GF by physical dispersion, which increases the specific surface area and active site of the electrode, thereby improving the catalytic performance of the electrode. The SEM images of the GF modified SnO2/rGO have been added into the manuscript.

  1. What about the surface developed by the rGO and rGO/SnO2? It is the reason for the increased current on glassy carbon modified by these materials. Surface area is often used in the discussion but no quantitative data are provided.

Reply: Thank you. It is known that rGO has large surface area and high conductivity. It is why that rGO has the intrinsic electrochemical catalysis for vanadium redox reaction, which offers large reaction place and fast electron transfer. Based on this, high-performance SnO2 is used to further improve the electrochemical catalysis of rGO. In this process, the amount of SnO2 nanoparticle is not so great to affect the surface area of rGO. We mainly study the effect of SnO2 on the catalysis of rGO for vanadium redox reaction.  

  1. Is missing some caracterization of the modified felt. Is it homogenous 1modification?

Reply: Thank you for your good advice. We have added SEM images of graphite felt and SnO2/rGO modified graphite felt to the manuscript.

  1. Pb with plot of impedance spectra that must show identical x and y axis.

Reply: Thank you for your good advice. The identical x and y axis in impedance spectra has been revised.

  1. It is truly graphene? It looks like relatively thick material on the SEM images.

Reply: Thank you. The home-made graphene oxide is applied in this work. The graphene oxide is directly observed by SEM without the special treatment such as ultrasonic dispersion. However, the XRD and TEM results are consistent with that reported in literatures, confirming that the material is true graphene oxide. The corresponding literature has been added into manuscript.

  1. What about the stability of this composite material in the conditions of VRFB?

Reply: Thank you. We conduct charge and discharge tests under different current densities. The performance of the modified cell is better than that of pristine one at each current density. It indicates that the composite material has good stability during the operation of cell.

  1. Overall, the authors claim for improved performances. They must improve the discussion about the mechanism that would allow any improvement compare to an activated GF. I do not expect a high impact of this work in the field of RFB.

Reply: Thank you. The SnO2/rGO can offers large surface area, more active sites and good conductivity for vanadium redox reactions, which is confirmed by electrochemical test results. The composite catalyst is used to modify graphite felt electrode. The high catalysis of SnO2/rGO can weaken the cell polarization including the electrochemical polarization and concentration polarization. It results in increase of electrolyte utilization, reduce of overpotential, and the improvement of energy density. The relevant mechanism has been added into manuscript. This work mainly is aimed at utilizing the synergistic effect of large surface area and high conductivity of carbon materials and high catalysis of metal oxide by an easy method and improving the energy storge performance of VRFB. In VRFB, many high-performance metal oxides have not been widely used in VRFB, due to their poor surface area and low conductivity. Therefore, the work can provide an easy method to improve the utilization of metal oxides in relevant electrochemical catalysis field.

However, if there are other errors or further requests, please contact me by e-mail. Thank you!

With sincere regards,

Mr. Zhangxing He

Reviewer 2 Report

Report on the manuscript “Bifunctional Synergistic Catalyst of Graphene/Sno2 for Vanadium Redox Reaction”, Y. Li et al.

 Ref. 1297894.

 General comments:

 The manuscript describes the preparation, characterization and electrocatalytic performance on the VO2+/VO2+ and V2+/V3+ couples of SnO2/reduced graphene oxide (SnO2/rGO) composite deposited onto GCE. The manuscript offers interesting results but incorporates several weaknesses so that major revision is recommended based on the following considerations.

General remarks:

 I) The title does not reflect the contents of the manuscript:

  1. a) “Reduced graphene oxide” rather than “graphene” is the carbonaceous material used.

  1. b) There is no unique “vanadium redox reaction”, the studied couples should be indicated.

  1. II) The authors study the electrocatalytic effects on the oxidation of VO2+ in 3M H2SO4 aqueous solution. In this regard:

  1. a) Fig. 4a should incorporate the voltammogram recorded at the bare GCE.

  1. b) The differences between the three tested electrode modifiers, SnO2, GO, and SnO2-rGO, in Fig. 4 should be taken with caution because the responses will depend on the net amount of electrode modifier transferred onto the basal GCE. In the Experimental section (page 3, section 2.3) the authors indicate that they start from 10 mg of catalyst, but it is unclear that “true” equivalent molar amounts of the different catalysts were used.

III) As is unfortunately frequent in recent literature, EIS data are treated in a simplistic way:

  1. a) The bias potential at which the EIS experiments were carried out is not indicated.

  1. b) The authors provide EIS data for a unique experiment. As a result, impedance data in Table 1 are provided with an unrealistic number of significant figures. The logical practice would be to use fitting data for replicate experiments and provide impedance parameters accompanied with the pertinent standard deviations.

  1. b) Impedance parameters are calculated without surface correction assuming that the surface area of all three modified electrodes, SnO2, GO, and SnO2-rGO, is the same. This is true in regard to the “apparent” geometrical area, but the effective area (taking into account the roughness, porosity, …) may be different.

  1. IV) The description of data for the V3+/V2+ couple is poor:

  1. a) Data on SnO2 for the V3+/V2+ couple are absent.

  1. b) What means “the negative electrolyte” in page 7, lines 232-233 and “negative redox reaction” in page 7, line 240?

  1. c) In page 7, lines 234-236 the authors write: “the reduction peak currents of both electrodes are greater than the oxidation peak currents. That is because the V3+ ions are more than V2+ ions in applied electrolyte”. This has no meaning; effectively, the authors use a solution of V3+ in 3M H2SO4, but in essentially reversible CVs such as in Figs. 4 and 5, the cathodic and anodic peaks slightly differ by the diffusion coefficients of the oxidized and reduced species. In fact, examination of Fig. 5a does not reveal significant differences at the SnO2-rGO electrode while the differences existing at the rGO (contrary to that in Fig. 4a, rGO rather than GO is apparently used here) has to be attributed, probably, to post-electron transfer reactions.

Author Response

Response to reviewers for Manuscript

Title: Synergistic catalysis of SnO2/reduced graphene oxide for VO2+/VO2+ and V2+/V3+ redox reactions

Corresponding Author: Mr. Zhangxing He

All Authors: Yongguang Liu, Yingqiao Jiang, Yanrong Lv, Zhangxing He, Lei Dai, Ling Wang.

Dear editor,

We must thank you and the reviewer for the hard work. Please do forward our heartfelt thanks to the expert for the valuable comments. Based on the comments and suggestions we received, we have made careful modifications and supplemented data on the previous manuscript. All changes are marked in red in the paper. We hope the revised manuscript will meet your magazine’s standard and the review’s requirements. Below you will find our point-by-point responses to the reviewer’s comments:

General comments:

The manuscript describes the preparation, characterization and electrocatalytic performance on the VO2+/VO2+ and V2+/V3+ couples of SnO2/reduced graphene oxide (SnO2/rGO) composite deposited onto GCE. The manuscript offers interesting results but incorporates several weaknesses so that major revision is recommended based on the following considerations.

General remarks:

  1. I) The title does not reflect the contents of the manuscript:
  2. a) “Reduced graphene oxide” rather than “graphene” is the carbonaceous material used.

Reply: Thank you for your good advice. We have changed the corresponding "graphene" to "reduced graphene oxide" in whole manuscript including the title.

  1. b) There is no unique “vanadium redox reaction”, the studied couples should be indicated.

Reply: Thank you for your advice. The vanadium redox reaction refers to the redox reaction of VO2+/VO2+ and V2+/V3+ in the title. It has been revised in title.

  1. II) The authors study the electrocatalytic effects on the oxidation of 1.6 M VO2+ in 3.0 M H2SO4 aqueous solution. In this regard:
  2. a) Fig. 4a should incorporate the voltammogram recorded at the bare GCE.

Reply: Thank you for your good advice. We have added the CV curve of the bare GCE to Fig. 4a.

  1. b) The differences between the three tested electrode modifiers, SnO2, GO, and SnO2/rGO, in Fig. 4 should be taken with caution because the responses will depend on the net amount of electrode modifier transferred onto the basal GCE. In the Experimental section (page 3, section 2.3) the authors indicate that they start from 10 mg of catalyst, but it is unclear that “true” equivalent molar amounts of the different catalysts were used.

Reply: Thank you. The preparation of the preliminary samples is based on the mass ratio. And, the preparation process causes quality loss. The relative molecular masses of reduced graphene oxide and SnO2 are very different. It is difficult to calculate the molar ratio. The mass ratio is mostly used in electrochemical catalysis such as evolution hydrogen and oxygen evolution reactions.

III) As is unfortunately frequent in recent literature, EIS data are treated in a simplistic way:

  1. a) The bias potential at which the EIS experiments were carried out is not indicated.

Reply: Thank you for your good advice. The bias potentials of the VO2+/VO2+ and V2+/V3+ reactions are 0.85 V and -0.45 V, respectively. We have added relevant content to the manuscript.

  1. b) The authors provide EIS data for a unique experiment. As a result, impedance data in Table 1 are provided with an unrealistic number of significant figures. The logical practice would be to use fitting data for replicate experiments and provide impedance parameters accompanied with the pertinent standard deviations.

Reply: Thank you. For EIS test, we mainly have two purposes. Firstly, we confirm the vanadium redox reactions under the control of the mass and charge transfer processes on the applied catalyst surface by the Nyquist plots. Then, for a quantizable comparison, a simplified equivalent circuit is employed to fit the experimental data. The obtained fitting data have been added into figure to replicate the experiments data. The fitting data can cover the experimental data well with a neglectable fitting deviation. The equivalent circuit is also a representative model for vanadium redox reaction in many literatures.

  1. c) Impedance parameters are calculated without surface correction assuming that the surface area of all three modified electrodes, SnO2, GO, and SnO2/rGO, is the same. This is true in regard to the “apparent” geometrical area, but the effective area (taking into account the roughness, porosity, …) may be different.

Reply: Thank you. The equal mass of catalyst is used to be studied the electrochemical activity. When the mass of the catalyst is equal, the catalysis improves as the effective area increases. Based the equations of  and , the difference in Rct reflects the change of electrochemical effective area (A). Therefore, it could be estimated that the electrochemical effective area of catalyst is in order of SnO2/rGO > GO > SnO2. The resistance per unit area (specific surface area) is not unsuitable for evaluation of catalysis here.

  1. IV) The description of data for the V3+/V2+ couple is poor:
  2. a) Data on SnO2 for the V3+/V2+ couple are absent.

Reply: Thank you. We aim at obtaining the composite SnO2/rGO by an in-situ method. The high-performance SnO2 is used to improve the catalysis of large-area GO for vanadium redox reaction. Firstly, it is confirmed that SnO2/rGO has the best catalysis for VO2+/VO2+ reaction than SnO2 and GO. Then, we verify that SnO2/rGO has better catalytic performance for V3+/V2+ couple than carbon support of GO.

  1. b) What means “the negative electrolyte” in page 7, lines 232-233 and “negative redox reaction” in page 7, line 240?

Reply: Thank you for your advice. “The negative electrolyte” is 1.6 M V3+ + 3.0 M H2SO4. We have revised descriptions in electrochemical measurement section. The "negative redox reaction" is the redox reaction of V3+ and V2+. They have been revised in manuscript.

  1. c) In page 7, lines 234-236 the authors write: “the reduction peak currents of both electrodes are greater than the oxidation peak currents. That is because the V3+ ions are more than V2+ ions in applied electrolyte”. This has no meaning; effectively, the authors use a solution of V3+ in 3 M H­2SO4, but in essentially reversible CVs such as in Figs. 4 and 5, the cathodic and anodic peaks slightly differ by the diffusion coefficients of the oxidized and reduced species. In fact, examination of Fig. 5a does not reveal significant differences at the SnO2-rGO electrode while the differences existing at the rGO (contrary to that in Fig. 4a, rGO rather than GO is apparently used here) has to be attributed, probably, to post-electron transfer reactions.

Reply: Thank you. For redox reaction, the cathodic and anodic peaks differ by many factors including diffusion coefficients of the oxidized and reduced species, concentration of active species, and electrochemical kinetics of oxidized and reduced species. However, the diffusion coefficient of vanadium ions is comparable. For V2+/V3+ redox reaction, the reduced V2+ ion is V3+ main solution. The high-concentration V3+ can affect the low-concentration V2+. In Fig. 5a, GO rather than rGO is used. The obvious difference exists in GO. It can be attributed to post-electron transfer due to more oxygen functional groups of GO than rGO. We have modified the relevant content.

However, if there are other errors or further requests, please contact me by e-mail. Thank you!

With sincere regards,

Mr. Zhangxing He

Round 2

Reviewer 1 Report

I would advice the authors to take into account all the comments that have been given in the first report:

Currently, EIS data are not very properly plotted, e.g. 20 ohm should have the same size in x and y axis. This is a very minor correction.

No information about change in porosity are provided. To claim is not enough. Surface area determination can be done with N2 adsorption or electrochemically. This latter method can certainly be performed rapidly with a selection of materials.

Stability means a bit longer cycling.

Author Response

Response to reviewers for Manuscript

Title: Synergistic catalysis of SnO2/reduced graphene oxide for VO2+/VO2+ and V2+/V3+ redox reactions

Corresponding Author: Mr. Zhangxing He

All Authors: Yongguang Liu, Yingqiao Jiang, Yanrong Lv, Zhangxing He, Lei Dai, Ling Wang.

Dear editor,

We must thank you and the reviewer for the hard work. Please do forward our heartfelt thanks to the expert for the valuable comments. Based on the comments and suggestions we received, we have made careful modifications and supplemented data on the previous manuscript. All revisions to the manuscript are marked up using the “Track Changes” function. All changes are marked in red in the paper. We hope the revised manuscript will meet your magazine’s standard and the review’s requirements. Below you will find our point-by-point responses to the reviewer’s comments:

Reviewer 1

General remarks:

I would advise the authors to take into account all the comments that have been given in the first report:

  1. Currently, EIS data are not very properly plotted, e.g., 20 ohm should have the same size in x and y axis. This is a very minor correction.

Reply: Thank you for your good advice. EIS data have been replotted with the same size in x and y axis. It is easy to see the diameter of semicircle and the slope of straight line.

2.No information about change in porosity are provided. To claim is not enough. Surface area determination can be done with N2 adsorption or electrochemically. This latter method can certainly be performed rapidly with a selection of materials.

Reply: Thank you for your good advice. The surface area of materials has been provided by N2 adsorption-desorption test.

  1. Stability means a bit longer cycling.

Reply: Thank you. In this work, we aimed at developing a composite catalyst of SnO2/rGO. The catalyst has large surface area, well conductivity and high activity. It can boost the electrochemical kinetics of electrode reaction and reduce the electrochemical polarization of cell. The energy efficiency of cell would be improved obviously at high current density.

However, if there are other errors or further requests, please contact me by e-mail. Thank you!

With sincere regards,

Mr. Zhangxing He

Reviewer 2 Report

The revised version of the manuscript is satisfactory. 

Author Response

Thank you very much for your approval of the revised manuscript.